# Sodium Alginate-Based MgO Nanoparticles Coupled Antibiotics as Safe and Effective Antimicrobial Candidates against *Staphylococcus aureus* of Houbara Bustard Birds

**DOI:** 10.3390/biomedicines11071959

**Published:** 2023-07-11

**Authors:** Maheen Murtaza, Amjad Islam Aqib, Shanza Rauf Khan, Afshan Muneer, Muhammad Muddassir Ali, Ahmad Waseem, Tean Zaheer, Lamya Ahmed Al-Keridis, Nawaf Alshammari, Mohd Saeed

**Affiliations:** 1Department of Zoology, Cholistan University of Veterinary and Animal Sciences, Bahawalpur 63100, Pakistan; maheenmurtaza152@gmail.com (M.M.); afshanchudhary9@gmail.com (A.M.); 2Department of Medicine, Cholistan University of Veterinary and Animal Sciences, Bahawalpur 63100, Pakistan; 3Department of Chemistry, University of Agriculture, Faisalabad 38000, Pakistan; shanzaraufkhan@gmail.com; 4Department of Zoology, Government Sadiq College Women University, Bahawalpur 63100, Pakistan; 5Institute of Biochemistry and Biotechnology, University of Veterinary and Animal Sciences, Lahore 54000, Pakistan; muddassir.ali@uvas.edu.pk; 6Houbara Foundation International, Lal Sohanra Park, Bahawalpur 63100, Pakistan; ahmaduvas98@gmail.com; 7Oryx Falcon Veterinarian, Doha 6763, Qatar; 8Department of Parasitology, University of Agriculture, Faisalabad 38000, Pakistan; teanzaheer942@gmail.com; 9Biology Department, Faculty of Science, Princess Nourah bint Abdulrahman University, Riyadh 11564, Saudi Arabia; 10Department of Biology, College of Science, University of Hail, Hail 55476, Saudi Arabia; naib.alshammari@uoh.edu.sa (N.A.); mo.saeed@uoh.edu.sa (M.S.)

**Keywords:** *S. aureus*, antibiotic resistance, MgO nanoparticles, sodium alginate, nanocomposites, antibacterial, genotoxicity

## Abstract

Alternative and modified therapeutic approaches are key elements in culminating antibiotic resistance. To this end, an experimental trial was conducted to determine the cytotoxicity and antibacterial potential of composites of magnesium oxide (MgO) nanoparticles and antibiotics stabilized in sodium alginate gel against multi-drug-resistant *Staphylococcus aureus* isolated from a houbara bustard. The characterization of preparations was carried out using X-ray diffraction (XRD), scanning transmissible electron microscopy (STEM), and Fourier-transform infrared spectroscopy (FTIR). The preparations used in this trial consisted of gel-stabilized MgO nanoparticles (MG), gel-stabilized tylosin (GT), gel-stabilized ampicillin (GA), gel-stabilized cefoxitin (GC), gel-stabilized MgO and tylosin (GMT), gel-stabilized MgO and cefoxitin (GMC), and gel-stabilized MgO and ampicillin (GMA). The study presents composites that cause a lesser extent of damage to DNA while significantly enhancing mitotic indices/phases compared to the other single component preparations with respect to the positive control (methyl methanesulphonate). It was also noted that there was a non-significant difference (*p* > 0.05) between the concentrations of composites and the negative control in the toxicity trial. Studying in parallel trials showed an increased prevalence, potential risk factors, and antibiotic resistance in *S. aureus*. The composites in a well diffusion trial showed the highest percentage increase in the zone of inhibition in the case of GT (58.42%), followed by GMT (46.15%), GC (40.65%), GMC (40%), GMA (28.72%), and GA (21.75%) compared to the antibiotics alone. A broth microdilution assay showed the lowest minimum inhibitory concentration (MIC) in the case of GMA (9.766 ± 00 µg/mL), followed by that of GT (13.02 ± 5.64 µg/mL), GMC (19.53 ± 0.00 µg/mL), GA (26.04 ± 11.28 µg/mL), GMT (26.04 ± 11.28 µg/mL), MG (39.06 ± 0.00 µg/mL), and GC (39.06 ± 0.00 µg/mL). The study thus concludes the effective tackling of multiple-drug-resistant *S. aureus* with sodium-alginate-stabilized MgO nanoparticles and antibiotics, whereas toxicity proved to be negligible for these composites.

## 1. Introduction

There are certain migratory birds that are affected by infections caused by antibiotic-resistant bacterial strains, leading to the loss of their lives. Houbara bustards (*Chlamydotis undulata*) are among the threatened species of migratory birds, and their population has decreased by between 30 and 50% [1] in recent years. Migratory birds travel long distances irrespective of geographical and political boundaries and are among the potential pathogen-transferring vehicles across borders [2]. Studies under focus should also include such overlooked areas that silently contribute to the dissemination of antibiotic resistance.

*S. aureus* and *E. coli* are becoming ubiquitous pathogens with variable strains of dairy [3,4,5] and pet animals in particular [6,7] all around the globe. Currently, a remarkable increase has been noticed in the emergence and re-emergence of antimicrobial-resistant strains of *Salmonella* and *E. coli* from birds too [8] in addition to vancomycin-resistant *S. aureus* [9] and methicillin-resistant *S. aureus* [10] from dairy animals. Such a situation is adding fuel to the fire regarding mass drug resistance. *S. aureus* has transformed from a commensal to a pathogenic bacterium by acquiring several antimicrobial-resistant genes. This pathogen is ubiquitous, can act as a zoonotic agent, and may also execute reverse zoonosis. A considerable amount of literature has been published about infections associated with *S. aureus* and its resistance against a wide range of antibiotics, making it a ubiquitous pathogen [11]. This situation is aggravated in lower- and middle-income countries like Pakistan [12], which require stern and comprehensive actions to leash drug resistance. Alternative approaches like nanoparticles, probiotics, prebiotics, and phytochemicals have been found to be effective approaches. The formers are as small as less than 100 nm in size in any dimension with proven antimicrobial properties. There is a wider list of both non-metallic and metallic nanoparticles that are being used as effective alternatives in various capacities against a wider range of pathogens.

Magnesium oxide nanoparticles (MgO NPs) possess characteristics that enable them to play a prominent role in biological and applied sciences. These are used as a biosensor for liver cancer assays [13], a tool for nano-cryosurgeries [14], an antimicrobial agent [15], and to treat conditions including heartburn [14]. The literature is still scarce on the cytotoxicity of these MgO NPs when applied in biomedical research; hence, it is imperative to evaluate the harmful impacts of MgO NPs at the cellular level that could warrant their safety to be used as a therapeutic agent [16,17].

There is a dire need to apply modified approaches/formulations that can reduce toxicity and maintain or enhance antibacterial activity. Sodium alginate is commonly used as a stabilizer in the food and pharmaceutical industries. The alginate was hypothesized to enhance the antibacterial activity of nanoparticles and antibiotics [18]. Nanoparticles exhibit unique physicochemical properties, such as high surface area-to-volume ratios and size-dependent reactivity. These properties can enhance the antimicrobial activity of nanoparticles against *S. aureus*. Nanoparticles can disrupt the bacterial cell wall, penetrate the bacterial membrane, and interfere with essential cellular processes, leading to the destruction of *S. aureus*.

The study of nanoparticles against *S. aureus* is important due to their potential to overcome antimicrobial resistance, enhance antimicrobial activity, enable targeted delivery, facilitate combination therapy, provide diagnostic capabilities, and improve wound healing. Further research in this field has the potential to revolutionize the treatment and management of *S. aureus* infections. The current research was planned to assess in vitro assessments of the antibacterial potential of antibiotics and nanoparticles separately stabilized sodium alginate, the combination of MgO nanoparticles and antibiotics stabilized in sodium alginate gel, and the cytotoxicity of nanoparticles.

## 2. Materials and Methods

### 2.1. Preparation of Nanoparticles and Composites

MgO nanoparticles were synthesized using hydrothermal technique in the presence of a surfactant. The weight by volume solution was prepared by dissolving 4 g of MgCl_2_.6H_2_O in 40 mL of distilled water. We followed the same protocol as in our previous study [18,19]. Sodium alginate 2% (*m*/*v*) and gelatin 2% (*m*/*v*) prepared in water were mixed in a ratio 4:1 (sodium alginate: gelatin) and homogenized at 500 rpm for 2 h using mechanical stirrer to prepare sodium alginate gel (G). The MgO nanoparticles (1.5 g) were added into 20 mL of gel and stirred for 4 h at 500 rpm to stabilize MgO within gel. Each drug (ampicillin, cefoxitin, and tylosine) solution (20 mL) was prepared by dissolving 0.035 g of drug in distilled water and mixing 20 mL of this solution in 20 mL of gel. The solution was further stirred for 4 h at 500 rpm and the final product was dried and ground to a fine powder. The composites formulated were as follows: MgO stabilized in gel = MG; MgO and tylosin stabilized in gel = GMT; MgO and cefoxitin stabilized in gel = GMC; MgO and ampicillin stabilized in gel = GMA; tylosin stabilized in gel = GT; ampicillin stabilized in gel = GA; and cefoxitin stabilized in gel = GC [18].

### 2.2. Characterization of Nanoparticles and Composites

Powder diffractometer Rigaku D/max Ultima III was used for X-ray diffraction (XRD) analysis of nanoparticles. It was operated at 40 kilo voltage (kV) and 0.130 ampere (A) current with a copper–potassium (Cu-Kα) radiation source emitting radiations with wavelengths of 0.15406 nm. Quanta 250, with operating voltage 30 kV, was used to obtain scanning electron microscopy (SEM) images of nanoparticles. Fourier-transform infrared spectroscopy (FTIR) was also applied to assess the characterization of products.

### 2.3. Part I: Cytogenetic Assessment of Different Treatments

#### 2.3.1. *Allium cepa* Ana-Telophase Test

The *Allium cepa* Ana-Telophase test was performed as per previous protocols [20,21] with minor modifications. Small-sized onions were kept in solutions of various concentrations for 48 h. Clearly, the following three distinct groups were made: (1) negative control onions kept in distilled water; (2) positive control onions kept in methyl methanesulfonate (MMS) (10 μg/mL); and (3) treated group onions kept in solutions consisting of gel-stabilized composites. The composites were further divided into MG, GMT, GMC, GMA, GA, GM, and GC to find their comparative cytotoxicity and genotoxicity. The composites were incubated with onion roots in the dark at room temperature for 24–48 h at concentrations of 1.25 mg/mL, 2.5 mg/mL, and 5 mg/mL. Root tips of onions were fixed with ethanol: acetic acid in a ratio of 3:1 in a *v/v* solution. After fixation, the root tips were washed with distilled water and fixed in 70% ethanol. For each treatment, 8 root tips were hydrolyzed for 10 min at 60 °C in 1N HCL and then rinsed in water. Root tips were stained with Schiff’s reagent for 30 min at room temperature. Darkly stained apical tips were taken and crushed on slides with 45% acetic acid. Then, these crushed slides along with cover slips were examined under microscope, and finally, microscopic pictures were photographed. To analyze mitotic activity under the effects of different treatments, 550 cells were counted [20,21]. The following formulas were used to evaluate the mitotic index (Equation (1)) and phase index (Equation (2)):(1)$$Mitotic\;index=\frac{Number\;of\;cells\;in\;division}{Number\;of\;total\;cells}\times 100\;$$
(2)$$Phase\;index=\frac{Particular\;phase\;}{Number\;of\;cells\;in\;division}\times 100$$

#### 2.3.2. Comet Assay on *A. cepa* Root Tips

The Comet test compared treated and control groups using root tips of onion bulbs (03 each). Root tips were crushed using nuclear isolation buffer (600 µL) with pH 7.5 for isolation of nuclei. The centrifugation was carried out at 4 °C for 7 min at 1200 rpm, and nuclear suspension was put on slides and coated with 1% normal melting point agarose (NMPA) at 37 °C. The slides were kept on ice for 5 min, which followed removal of covers slips and immersion of slides in electrophoresis tank with fresh electrophoresis buffer for 20 min. The immersions were carried out at 300 mA for 20 min at 25 V. The staining of slides was carried out for 5 min in dark with 20 μg/mL ethidium bromide, following which the cover slips were placed on slides. Three slides from each sample were analyzed with BAB TAM-F fluorescence microscope. DNA damage was qualitatively classified as ranging from 0 to 4, depending on head integrity and tail length [20,21]. For each sample, the following formula was used to calculate total DNA damage in arbitrary units (Equation (3)):(3)$$Arbitrary\;Unit=\sum_{i=0}^{4}Ni\times i\;$$

*Ni* = Number of cells.

*i* = degree of damage (0–4).

### 2.4. Part II: Antimicrobial Potential against Bacteria

#### 2.4.1. Isolation of *S. aureus*

The antibacterial potential of composites was evaluated by selecting *S. aureus* from migratory bird houbara bustards (*C. undulata*). The selected birds were kept in captivity at the Houbara International Foundation, Lal Sohanra Park, Bahawalpur. The selection of this bird was based on its migratory history as well as captivity in a controlled but natural environment. Cloacal samples (*n* = 105) adopting purposive sampling protocol were taken from birds early in the morning at repeated and feasible time intervals using sterile swabs [22]. The samples (kept at 4 °C) were transported to the Central Diagnostic Laboratory, Cholistan University of Veterinary and Animal Sciences, Bahawalpur, for further processing. The samples were processed for biochemical tests as per defined protocol for identification of *S. aureus* [7,23].

#### 2.4.2. Molecular Analysis

The molecular assay was adopted to confirm *S. aureus* by targeting *Nuc* gene with forward primers 5′AAGGGCAATACGCAAAGAG 3′ and *Nuc* reverse primers 5′AAACATAAGCAACTTTAGCCAAG 3′. An amount of 20 μL of reaction volume was prepared using 10 μL of PCR 2× master mix (Thermo scientific Catalog # K0171, Waltham, MA, USA), 1 μL of forward primer (10 pmoL), 1 μL of reverse primer (10 pmoL), 2 μL of DNA (50 ng/L), and 6 μL of deionized water. The thermocycler profile comprised initial denaturation at 94 °C (5 min), denaturation at 94 °C (45 s), annealing at 63–53 °C (45 s), extension at 72 °C (45 s), and final extension at 72 °C (45 s) for 35 cycles.

#### 2.4.3. Antibacterial Potential of Nanocomposites

Multi-drug-resistant *S. aureus* (resistant to more than 2 classes of antibiotics) was selected for trial of assessing antibacterial potential of composites. Both well diffusion and broth microdilution methods were applied to validate antibacterial activity and to find minimum effective doses of different composites at frequent intervals of incubation.

##### Well Diffusion Assay

Antibacterial activity in terms of zone of inhibition (mm) was measured by preparing wells (8 mm in diameter) in sterile Mueller–Hinton agar and subsequently adding preparations at the rate of 0.01 mg/mL, following spreading the fresh culture of bacteria adjusted at 1–1.5 × 10^8^ CFU/mL (equal to 0.5 McFarland). With the help of vernier calipers, zone of inhibition produced around wells was measured after incubation of 24 h at 37 °C [24].

##### Broth Microdilution Assay

Minimum inhibitory concentration of composites at different time intervals was assessed using broth microdilution assay. To briefly describe the protocol, sterile nutrient broth (50 μL) was added in all wells, followed by two-fold dilution of composites starting from 10 mg/mL in all wells except in positive control. The positive control contained broth and fresh culture, while negative control only contained broth. The fresh growth of bacteria adjusted at 1.5 × 10^5^ CFU/mL (50 μL) was added to all wells except the one designated as negative control. The plates were incubated for 20 h at 37 °C. Optical density (OD) values before and after incubation were taken at a wavelength of 690 nm to determine inhibition of bacterial growth [24]. The OD values were also taken at 4, 8, 12, 16, 20, and 24th h of incubation to compare the effects of different time intervals on antibacterial potential of composites against multiple-drug-resistant *S. aureus*.

### 2.5. Statistical Analyses

The data collected were analyzed by both parametric and non-parametric tests. While *t*-test and ANOVA were applied to data from two groups and that from more than two groups, respectively. Minitab (Version 17, Brandon Court, Unit E1-E2 Progress Way, Coventry, UK) and SPSS (Version 22, IBM Corp., Armonk, NY, USA) for data analysis were used, and the significance of the data was decided on *p* < 0.05.

## 3. Results

### 3.1. Characterization of Nanoparticles and Composites

The XRD patterns of MgO nanoparticles (Figure 1A) were obtained after calcination at 400 °C, while narrower diffraction peaks at higher temperatures confirmed the formation of MgO nanoparticles. Miller indices indicated on the peaks (Figure 1A) measured 111, 200, 220, 31,1, and 222 at 2-theta 43.0°, 46.0°, 63.0°, 75.0°, and 78.0°, respectively (ICDD card no. 77–2364). Data from micrographs showed that it was a Face-Centered Cubic (FCC) structured and space group Fm-3 m (structural parameters, Appendix A). Peaks at 25°, 38°, and 66° 2-theta values were indexed to (001), (101), and (103), respectively (ICDD card no. 84–2163). This set represented the presence of traces of Mg(OH)_2_ in the products, as the intensity of these three peaks was very low compared to the other peaks. Low and intense peaks were not identified because of the noise. The morphology of nanoparticles using scanning electron microscopy (SEM) revealed that the size of the nanoparticles ranged from 80 to 200 nm, approximately (Figure 1B). Strong bands were observed for scomposites at 1500–1600 cm^−1^, representing the presence of the carbonyl functional group in GMC. Due to the presence of amine (NH_2_ and NH) groups, two peaks were observed around 3300–3600 cm^−1^. In the case of MG, amine groups were absent. Only hydroxyl groups were present. So, a broad band was observed at around 3300–3600 cm^−1^. MgO peaks were present before 1000 cm^−1^. A comparison of both patterns confirmed that the drug had been coated because the characteristic peaks of amine were obtained.

### 3.2. Cytotoxicity and Genotoxicity of Different Preparations

The impact of antibiotics alone, sodium-alginate-stabilized nanoparticles, sodium-alginate-stabilized antibiotics, and both nanoparticles and antibiotics stabilized in sodium alginate on mitotic index (MI) and phase index on roots of *A. cepa* and their impacts on DNA damage at various concentrations were analyzed. Nanoparticles and antibiotics stabilized in sodium alginate showed non-significant (*p* < 0.05) responses compared with those of the negative control, which reflects their safe use on host species like humans and animals.

#### 3.2.1. Effect of Cefoxitin

The cytotoxic and genotoxic effects were analyzed as a measure of the reduction in the mitotic index and mitotic phases following the application of cefoxitin (Table 1). DNA damage was found to be in direct proportion with the time and concentration of cefoxitin (Figure 2 and Figure 3).

#### 3.2.2. Effect of Magnesium Oxide (M) and Gel (G)

Significant cytotoxic and genotoxic impacts of the MG were observed on onion root cells (*p* < 0.05). Time- and concentration-dependent decreases in MI and mitotic phases were shown by the MG (Table 2).

#### 3.2.3. Effect of Gel, Magnesium Oxide, and Cefoxitin (GMC)

It was observed from the study that there were non-significant effects of cytotoxicity and genotoxicity at various concentrations compared with those of the negative control. Time- and concentration-dependent increases in the mitotic index and mitotic phases were observed compared to the control. Similarly, a decrease in DNA damage was noted in a time- and concentration-dependent manner under the effect of GMC on the root tips of onions compared to the control (Table 3).

#### 3.2.4. Effect of Cefoxitin and Gel (GC)

Our study found non-significant (*p* > 0.05) cytotoxic and genotoxic effects from the application of GC compared to those of the negative control group (Table 4). It was also observed that increases in the mitotic index and mitotic phases and decreases in DNA damage under the effect of GC were also time- and concentration-dependent.

#### 3.2.5. Effect of Tylosin and Gel (GT)

The observations regarding the cytotoxic and genotoxic effects of GT on the onion root tips were non-significant compared to the negative control. It was also found that increases in mitotic index and mitotic phases and decreases in DNA damage under the effect of GC were also time- and concentration-dependent. Similarly, decreases in DNA damage were observed compared to the positive control (Table 5).

#### 3.2.6. Effect of Ampicillin and Gel (GA)

This study explored the non-significant (*p* > 0.05) cytotoxic and genotoxic effects when GA was evaluated and compared with the negative control. On the other hand, increases in the mitotic index and mitotic phases in *A. cepa* cells were both time- and concentration-dependent. A reduction in DNA damage was also observed by ampicillin and gel on onion root tips (Table 6).

#### 3.2.7. Effect of Gel, Magnesium Oxide, and Tylosin (GMT)

No significant cytotoxic and genotoxic effects were observed by GMT treatment. Here, in the case of GMT (gel, magnesium oxide, and tylosin) again, increases the in mitotic index and mitotic phases followed a time- and concentration-dependent strategy. Similarly, a reduction in DNA damage was noted on onion root tips in the case of GMT (Table 7).

### 3.3. Antibacterial Potential of Composites against Bacteria

#### 3.3.1. Comparison of the Zones of Inhibition

The isolated bacteria following biochemical characterization were confirmed as *S. aureus* through *nuc gene* (Figure 4). The isolates positive for both biochemical and molecular assay were put to further study against different preparations.

The current study has reported a significant difference (*p* < 0.05) between composites (nanoparticles and antibiotics stabilized in sodium alginate) in comparison with the preparations used alone (Table 8). The current study noted 58.42% and 46.15% increases in the ZOI in cases of GT and GMT, respectively, compared to tylosin alone. Increases in the ZOI of GC and GMC were 40.65 and 40%, respectively, compared to cefoxitin alone. The difference between composites with tylosin alone was found to be significantly different (*p* < 0.05), and the same was noted in a comparison of cefoxitin-based composites compared to cefoxitin alone. The composites GA, GMA, and MA presented 21.75, 28.72%, and −2.87% variations in ZOI compared to ampicillin alone. Comparison of composites of MgO nanoparticles GMC, GMT, GMA, and MG showed 25.32, 28.19, 44.55, and 17.65% increases in ZOI compared to that of MgO alone (M).

#### 3.3.2. Comparison of Minimum Inhibitory Concentrations (MIC)

##### Antibacterial Efficacy of Composites with Respect to Time Intervals

The potential of each composite to show antibacterial activity with respect to the incubation period was found to be significantly different (*p* < 0.05) at various hours of incubation (Figure 5). A significant reduction in MIC in the case of GMA was noticed at the 8th hour of incubation, while it was further significantly reduced at the 16th hour of incubation (*p* < 0.05), which thereafter remained non-significant (*p* > 0.05). This trend showed that GMA could be used effectively for its maximum efficacy at the 16th hour of incubation, while an early response could be obtained at the 8th hour of incubation. GA and GT showed significant reductions (*p* < 0.05) in MIC at the 12th hour of incubation, which remained non-significant (*p* > 0.05) onward. GMC presented a significant reduction (*p* > 0.05) in MIC at the 8th hour of incubation, which was further reduced significantly (*p* < 0.05) at the 20th hour, but onward, there was a non-significant difference (*p* > 0.05). GMT and MG showed a significant difference (*p* < 0.05) in MICs at all the hours of incubation periods, indicating a wider range of the antibacterial potential of these composites.

##### Comparison of the Antibacterial Potential among Different Composites

A comparison of the different composites at each incubation period showed significant (*p* < 0.05) results (Table 9). Composites consisting of both antibiotics and nanoparticles stabilized in sodium alginate gel were found to be more effective than those of antibiotics or nanoparticles alone. At the fourth hour of incubation, the highest MIC (1042 ± 361 µg/mL) against *S. aureus* was noted in the cases of GT and MG composite, followed by GA, GC, GMA, GMC, and GMT. At the eighth hour of incubation, the highest MIC (833 ± 361 µg/mL) against *S. aureus* was noted in the case of GT composites, followed by GA, GC, MG, GMA, GMC, and GMT. At the 12th hour of incubation, the highest MIC (521 ± 180 µg/mL) against *S. aureus* was noted in the case of GT composites, followed by GA, MG, GMA, GC, GMC, and GMT. At the 16th hour of incubation, the highest MIC (260.4 ± 90.2 µg/mL) against *S. aureus* was noted in the case of GC composites, followed by MG, GA, GMC, GT, GMT, and GMA. At the 20th hour of incubation, the highest MIC (156.3 ± 0.0 µg/mL) against *S. aureus* was noted in the case of GC, followed by MG, GA, GT, GMT, GMC, and GMA. At the 24th hour of incubation, the highest MIC (39.06 ± 0.00) against *S. aureus* was noted in the case of MG and GC composites, followed by GA, GMT, GMC, GT, and GMA. The outcome of the comparison of different composites revealed nanoparticles and antibiotics stabilized in gel to be the best therapeutics against multiple-drug-resistant bacteria.

## 4. Discussion

### 4.1. Characterization of Nanoparticles

The spherical form and smooth surface of MgO nanoparticles in the current study were in line with previous studies [18,19]. Our findings of the clumping of some particles and some being well scattered were also in line with previous studies. The average size of nanoparticles was 16 nm, while the range of 7–38 revealed the spherical shape of nanoparticles. For the production of the desired nanoparticles, repaid reduction, assembly, and sintering at room temperature to the spherical shape were carried out [25].

### 4.2. Genotoxicity Assay

The results of DNA damage in *A. cepa* root tips were in line with those of previous studies [26,27], where non-significant differences were observed between the positive control and that of treatment with 100 µg/L clopyralid for all incubation periods except 24 h. The gradual increase in the CAs reveals the genotoxic effects of clopyralid. It was also found in a study [27] that decreases in the mitotic index were concentration-dependent decreases in MI (r = −0.99) at all concentrations of WO_3_NPs compared to those of the negative control group. The negative control expressed the highest mitotic index (MI) value, whereas the highest concentration of WO_3_NPs in their study revealed the lowest MI value (24.64 ± 0.72). It was also noted from their study that decreases in MI were noted sooner after 12.5 mg/L than in the positive control. Following the exposure of WO_3_NPs, a dose-dependent increase in the mitotic phases was noted at all concentrations compared to that of the negative control group but otherwise in the prophase.

### 4.3. Antibacterial Potential Nanocomposites

In the present study, composites GA, GMA, and MA presented 21.75, 28.72%, and −2.87% variations in the ZOI compared to ampicillin alone, while a comparison of composites of MgO nanoparticles GMC, GMT, GMA, and MG showed 25.32, 28.19, 44.55, and 17.65% increases in the ZOI compared to that of MgO alone (Table 9). In several studies, MgO nanoparticles have demonstrated promising potential as an antibacterial agent against bacteria, where growth was reduced to >95% at higher dosages (>5 mg/mL) [15]. Ampicillin in combination with silver nanoparticles has been reported to exhibit a higher efficacy at lower concentrations [28]. It was also reported that supplementing rations with ZnO and Copper oxide nanoparticles significantly reduced growth of bacteria and promoted removal of resistant genes [29]. The antibacterial efficacy of sodium alginate/gelatin films with propolis showed a 0.338 mg/mL MIC and that the growth of *S. aureus* was significantly reduced [30]. Metallic oxide nanoparticles on the other hand also significantly inhibited growth of bacteria like *Streptococcus* and *Klebsiella* [31]. On the other hand, Zn-, Cu-, and Mg-based composites were not found to be effective at stopping bacterial growth. This phenomenon was explained by the fact that the alteration in the release of ions resulted in an altered biocompatibility of metals, and hence there was a change in the antibacterial activity. However, the addition of Mg to Ag in nanocomposites gave a boost to the antibacterial activity, which was due to the increased quantity and rate of release of silver ions. The study of [32] reported unique physiochemical properties of metallic nanoparticles, which express significant antibacterial activity, while their toxicities varied depending upon structure, shape, dimension, and size. MgO nanoparticles were found to have a wider range of applications due to their chemical stability and activity [33]. They also studied the in vivo responses of MgO nanoparticles in lab animals, where a higher percentage of tail DNA in tissues of liver cells was noticed in response to 500 mg/kg. However, the safety data and impacts on humans’ health are yet to be determined.

The rising resistance to the antibiotics by *S. aureus* poses serious concerns not only for animals but also for public health as it produces often the suppurative anomalies which are associated with severe tissue damage and, finally, necrosis [34]. Hence use of alternative to antimicrobials like nanoparticles can be a productive shift towards effectiveness of therapeutics. It is also noteworthy that mere using extracts of plants may not serve the purpose of modulation of resistance but if nanoparticles are prepared from these plant extracts, the sustained and safe antimicrobial alternatives may be executed [35,36]

## 5. Conclusions

Gel-stabilized composites of MgO nanoparticles and antibiotics (particularly ampicillin) proved to be better potential antibacterial candidates than MgO alone or antibiotics alone. The antibacterial contact time for the composites was found to reflect a quick response in the early hours of incubation. Cytotoxicity and genotoxicity trials of MgO nanoparticles composites and the antibiotics stabilized in sodium alginate gels proved these to be safe for use in biomedical research. This study thus concludes that gel-based composites of nanoparticles and antibiotics are potential antibacterial candidates and are lower in toxicity, which calls for the development of therapeutic regimens through in vivo and field trials. Extensive studies are required to validate the outcomes of therapeutic and toxicity trials; the refinement of dose regimens is another challenge, as is the binding ability of drugs with nanoparticles and the stability of nanoparticle structure while using in vivo trials. However, these trials may strengthen the strategies to counter antimicrobial resistance and infection at a satisfactory level.

## Figures and Tables

**Figure 1 biomedicines-11-01959-f001:**
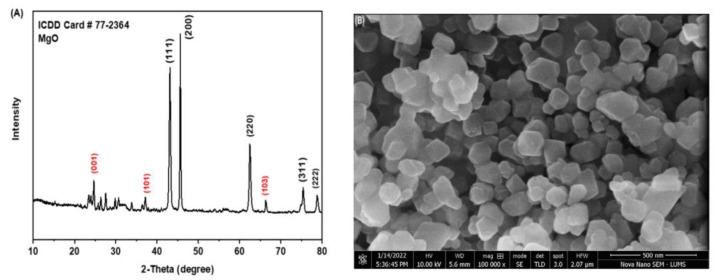
XRD and SEM images of MgO nanoparticles. (**A**) XRD image (XRD pattern of synthesized MgO nanoparticles); (**B**) SEM image of MgO nanoparticle.

**Figure 2 biomedicines-11-01959-f002:**
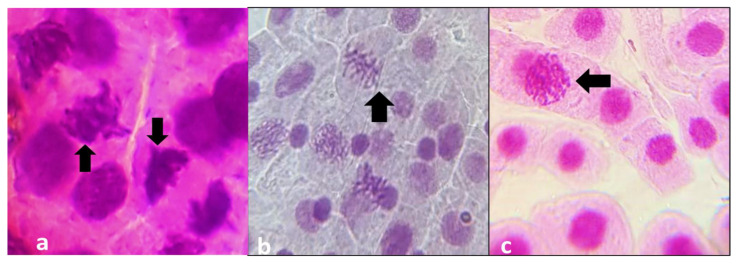
Different stages of mitosis in onion root meristematic cells: Arrows show in (**a**) stickiness of chromosomes; (**b**) metaphase abnormalities; (**c**) normal prophase stage.

**Figure 3 biomedicines-11-01959-f003:**
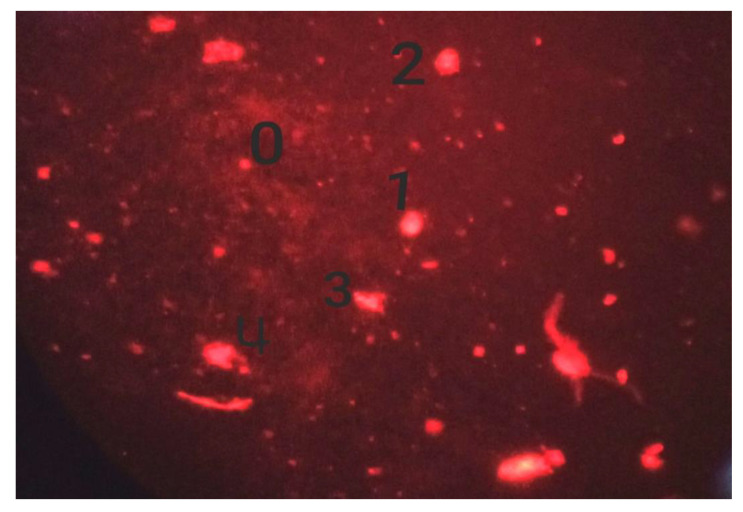
Induced DNA damage, where 0 = no damage; 1 = mild damage; 2 = moderate damage; 3 = severe damage; and 4 = complete DNA damage.

**Figure 4 biomedicines-11-01959-f004:**
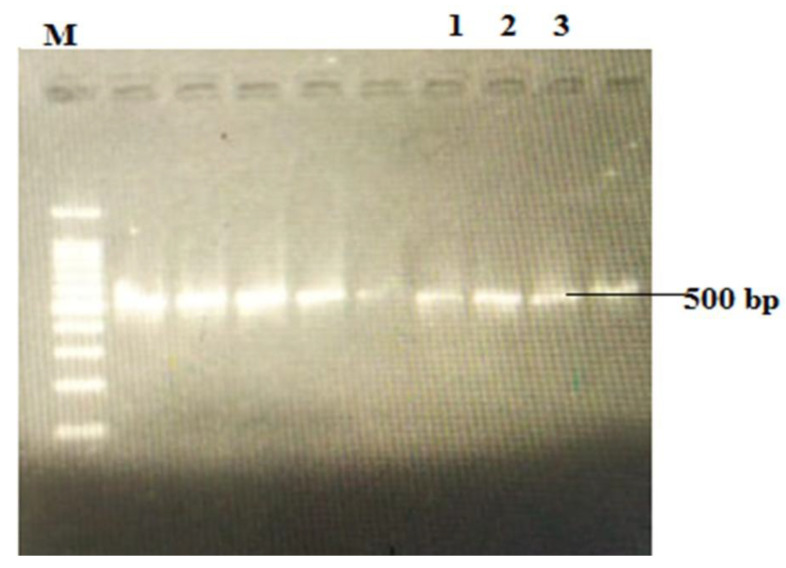
PCR amplicons of *nuc* gene (*S. aureus*). M = 1 kb DNA ladder (Geneon 1 kb DNA ladder); Lane 1: Staph-1; Lane 2: Staph-2; and Lane 3: Staph-3.

**Figure 5 biomedicines-11-01959-f005:**
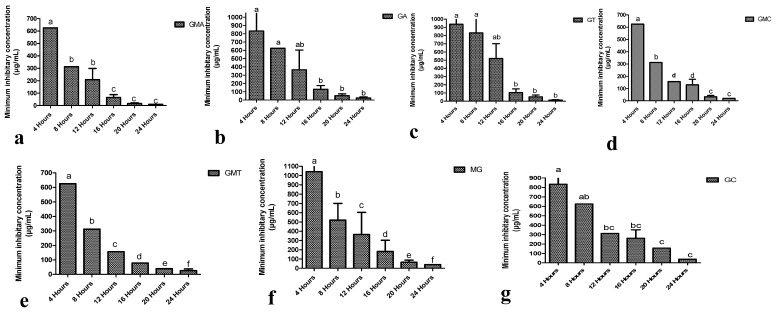
Composites’ antibacterial response at different time intervals of incubation (hours): (**a**) Gel, MgO, and ampicillin (GMA); (**b**) gel and ampicillin (GA); (**c**) gel and tylosin (GT); (**d**) gel, MgO, and cefoxitin (GMC); (**e**) gel, MgO, and tylosin (GMT); (**f**) MgO and gel (MG); (**g**) gel and cefoxitin (GC)**.** Different letters among different hours for each preparation show the statistically significant difference (*p* ≤ 0.05).

**Table 1 biomedicines-11-01959-t001:** Effect of cefoxitin on mitotic, phase index, and DNA damage in *A. cepa* roots at different concentrations.

Concentration (mg/mL)	CCN	MI ± SD	Phase Index (%) ± SD				DNA Damage (Mean ± SD)
			Prophase	Metaphase	Anaphase	Telophase	
**24 h**							
Control	508	78.22 ± 0.12 ^a^	98.09 ± 0.72 ^a^	2.02 ± 0.12 ^a^	2.62 ± 0.28 ^a^	7.98 ± 0.19 ^a^	12 ± 0.99 ^a^
MMS	509	56.15 ± 0.67 ^b^	84.13 ± 0.06 _b_	2.01 ± 0.16 ^b^	1.2 ± 0.01 ^b^	5.88 ± 0.87 ^b^	122 ± 0.75 ^b^
1.25 mg/mL Cefoxitin	550	52.02 ± 0.11 ^b^	84.09 ± 0.11 ^b^	2.11 ± 0.02 ^b^	1.99 ± 1.16 ^b^	5.75 ± 0.12 ^b^	105 ± 0.99 ^b^
2.5 mg/mL Cefoxitin	567	50.19 ± 0.28 ^b^	83.12 ± 0.02 ^b^	1.99 ± 0.02 ^b^	1.90 ± 0.01 ^b^	5.23 ± 0.05 ^b^	107 ± 0.19 ^b^
5 mg/mL Cefoxitin	565	49.03 ± 0.92 ^b^	82.33 ± 0.41 ^b^	1.86 ± 0.13 ^b^	1.87 ± 0.28 ^b^	4.25 ± 0.14 ^b^	109 ± 2.11 ^b^
**48 h**							
1.25 mg/mL Cefoxitin	567	50.32 ± 0.29 ^b^	83.09 ± 0.01 ^b^	1.85 ± 0.05 ^b^	1.89 ± 1.26 ^b^	5.55 ± 0.11 ^b^	116 ± 0.32 ^b^
2.5 mg/mL Cefoxitin	555	49.11 ± 0.22 ^b^	82.12 ± 0.12 ^b^	1.59 ± 0.01 ^b^	1.70 ± 0.11 ^b^	5.33 ± 0.13 ^b^	126 ± 0.21 ^b^
5 mg/mL Cefoxitin	545	48.26 ± 0.93 ^b^	82.03 ± 0.11 ^b^	1.56 ± 0.18 ^b^	1.77 ± 0.18 ^b^	5.13 ± 0.01 ^b^	120 ± 0.21 ^b^

Different letters in the same columns show the statistically significant difference (*p* ≤ 0.05) among treatment groups. MMS = Methyl methanesulphonate; CCN = counting cell numbers; SD = Standard deviation.

**Table 2 biomedicines-11-01959-t002:** Effect of magnesium oxide (M) and gel (G) on mitotic, phase index, and DNA damage in *A. cepa* roots at various concentrations.

Concentration (mg/mL)	CCN	MI ± SD	Phase Index (%) ± SD				DNA Damage (Mean ± SD)
			Prophase	Metaphase	Anaphase	Telophase	
**24 h**							
Control	508	78.22 ± 0.12 ^a^	98.09 ± 0.72 ^a^	2.02 ± 0.12 ^a^	2.62 ± 0.28 ^a^	7.98 ± 0.19 ^a^	12 ± 0.99 ^a^
MMS	509	56.15 ± 0.67 ^b^	84.13 ± 0.02 ^b^	2.01 ± 0.16 ^b^	1.2 ± 0.01 ^b^	5.88 ± 0.87 ^b^	122 ± 0.75 ^b^
1.25 mg/mL MG	550	51.21 ± 0.01 ^b^	83.09 ± 0.12 ^b^	2.25 ± 0.12 ^b^	1.89 ± 1.26 ^b^	5.71 ± 0.22 ^b^	105 ± 0.99 ^b^
2.5 mg/mL MG	567	50.18 ± 0.32 ^b^	82.12 ± 0.12 ^b^	2.09 ± 0.01 ^b^	1.80 ± 0.22 ^b^	5.13 ± 0.15 ^b^	107 ± 0.19 ^b^
5 mg/mL MG	565	49.13 ± 0.28 ^b^	81.33 ± 0.11 ^b^	1.76 ± 0.11 ^b^	1.17 ± 0.19 ^b^	4.25 ± 0.15 ^b^	109 ± 2.11 ^b^
**48 h**							
1.25 mg/mL MG	567	49.12 ± 0.24 ^b^	82.09 ± 0.21 ^b^	1.75 ± 0.03 ^b^	1.99 ± 1.46 ^b^	5.15 ± 0.45 ^b^	116 ± 0.32 ^b^
2.5 mg/mL MG	555	49.99 ± 0.02 ^b^	81.12 ± 0.32 ^b^	1.19 ± 0.04 ^b^	1.86 ± 0.31 ^b^	4.23 ± 0.56 ^b^	126 ± 0.21 ^b^
5 mg/mL MG	545	48.16 ± 0.09 ^b^	80.03 ± 0.99 ^b^	1.66 ± 0.11 ^b^	1.77 ± 0.17 ^b^	4.83 ± 0.21 ^b^	120 ± 0.21 ^s^

Different letters in the same columns show the statistically significant among treatment groups (*p* ≤ 0.05). MMS = Methyl methanesulphonate; CCN = counting cell numbers; MG = MgO nanoparticles (M) stabilized in sodium alginate gel (G); SD = Standard deviation.

**Table 3 biomedicines-11-01959-t003:** Effect of gel, magnesium oxide, and cefoxitin (GMC) on mitotic and phase index in *A. cepa* roots. Effect of cefoxitin and gel on DNA damage in *A. cepa* root tips at different concentrations.

Concentration (mg/mL)	CCN	MI ± SD	Phase Index (%) ± SD				DNA Damage (Mean ± SD)
			Prophase	Metaphase	Anaphase	Telophase	
**24 h**							
Control	586	78.22 ± 0.12 ^a^	98.09 ± 0.72 ^a^	2.02 ± 0.12 ^a^	2.62 ± 0.28 ^a^	7.98 ± 0.19 ^a^	12 ± 0.99 ^a^
MMS	575	56.15 ± 0.67 ^b^	84.13 ± 0.06 ^b^	2.01 ± 0.16 ^b^	1.2 ± 0.01 ^b^	5.88 ± 0.87 ^b^	122 ± 0.75 ^b^
1.25 mg/mL GMC	577	66.11 ± 0.11 ^a^	89.13 ± 0.23 ^a^	2.1 ± 0.12 ^a^	1.98 ± 0.16 ^a^	6.64 ± 0.90 ^a^	75 ± 0.19 ^a^
2.5 mg/mL	550	63.19 ± 0.12 ^a^	88.19 ± 0.45 ^a^	2.02 ± 0.03 ^a^	1.78 ± 0.21 ^a^	6.33 ± 0.85 ^a^	85 ± 0.09 ^a^
5 mg/mL	545	60.22 ± 0.12 ^a^	86.10 ± 0.11 ^a^	2.07 ± 0.18 ^a^	1.77 ± 0.22 ^a^	6.29 ± 0.21 ^a^	87 ± 1.02 ^a^
**48 h**							
1.25 mg/mL	545	67.19 ± 0.01 ^a^	87.13 ± 0.13 ^a^	2.55 ± 0.12 ^a^	1.88 ± 0.18 ^a^	6.69 ± 0.09 ^a^	76 ± 0.06 ^a^
2.5 mg/mL GMC	559	65.09 ± 0.02 ^a^	86.19 ± 0.05 ^a^	2.12 ± 0.13 ^a^	1.68 ± 0.11 ^a^	6.03 ± 0.05 ^a^	82 ± 0.07 ^a^
5 mg/mL	550	62.12 ± 0.13 ^a^	86.98 ± 0.23 ^a^	2.08 ± 0.13 ^a^	1.57 ± 0.20 ^a^	6.09 ± 0.71 ^a^	85 ± 0.15 ^a^

Different letters in the same columns show the statistically significant among treatment groups (*p* ≤ 0.05). MMS = Methyl methanesulphonate; CCN = counting cell numbers; SD = Standard deviation.

**Table 4 biomedicines-11-01959-t004:** Effect of cefoxitin and gel (GC) on mitotic and phase index in *A. cepa* roots. Effect of cefoxitin and gel on DNA damage in *A. cepa* root tips at different concentrations.

Concentration (mg/mL)	CCN		Phase Index (%) ± SD				DNA Damage (Mean ± SD)
			Prophase	Metaphase	Anaphase	Telophase	
**24 h**							
Control	586	78.22 ± 0.12 ^a^	98.09 ± 0.72 ^a^	2.02 ± 0.12 ^a^	2.62 ± 0.28 ^a^	7.98 ± 0.19 ^a^	12 ± 0.99 ^a^
MMS	575	56.15 ± 0.67 ^b^	84.13 ± 0.06 ^b^	2.01 ± 0.16 ^b^	1.2 ± 0.01 ^b^	5.88 ± 0.87 ^b^	122 ± 0.75 ^b^
1.25 mg/mL GC	577	65.11 ± 0.31 ^a^	88.19 ± 0.27 ^a^	2.99 ± 0.12 ^a^	1.78 ± 0.16 ^a^	6.94 ± 0.17 ^a^	64 ± 0.49 ^a^
2.5 mg/mL GC	550	63.18 ± 0.02 ^a^	87.14 ± 0.49 ^a^	2.12 ± 0.23 ^a^	1.71 ± 0.21 ^a^	6.23 ± 0.15 ^a^	72 ± 0.68 ^a^
5 mg/mL GC	545	62.28 ± 0.42 ^a^	86.02 ± 0.17 ^a^	2.99 ± 0.22 ^a^	1.08 ± 0.22 ^a^	5.99 ± 0.22 ^a^	72 ± 1.02 ^a^
**48 h**							
1.25 mg/mL GC	545	66.89 ± 0.01 ^a^	87.99 ± 0.03 ^a^	2.95 ± 0.19 ^a^	1.68 ± 0.08 ^a^	6.69 ± 0.09 ^a^	79 ± 0.06 ^a^
2.5 mg/mL GC	559	65.19 ± 0.22 ^a^	85.09 ± 0.04 ^a^	2.82 ± 0.53 ^a^	1.44 ± 0.16 ^a^	6.03 ± 0.02 ^a^	80 ± 0.97 ^a^
5 mg/mL GC	555	64.99 ± 0.09 ^a^	84.11 ± 0.04 ^a^	2.92 ± 0.73 ^a^	1.32 ± 0.19 ^a^	5.923 ± 0.15 ^a^	80 ± 0.85 ^a^

Different letters in the same columns show the statistically significant among treatment groups (*p* ≤ 0.05). MMS = Methyl methanesulphonate; CCN = counting cell numbers; SD = Standard deviation.

**Table 5 biomedicines-11-01959-t005:** Effect of tylosin and gel (GT) on mitotic and phase index in *A. cepa* roots and effect on DNA damage in *A. cepa* root tips at different concentrations.

Concentration (ppm)	CCN	MI ± SD	Phase Index (%) ± SD				DNA Damage (Mean ± SD)
			Prophase	Metaphase	Anaphase	Telophase	
Control	5006	76.81 ± 0.92 ^a^	99.18 ± 0.61 ^a^	2.19 ± 0.21 ^a^	2.61 ± 0.11 ^a^	7.81 ± 0.88 ^a^	12 ± 0.99 ^a^
MMS	5704	59.15 ± 0.67 ^b^	86.13 ± 0.46 ^b^	2.01 ± 0.26 ^b^	1.3 ± 0.11 ^b^	6.15 ± 0.15 ^b^	122 ± 0.75 ^b^
**24 h**							
1.25 mg/mL GT	504	75.25 ± 0.75 ^a^	95.43 ± 0.46 ^a^	2.13 ± 0.16 ^a^	2.09 ± 0.12 ^a^	7.85 ± 0.55 ^a^	22 ± 0.19 ^a^
2.5 mg/mL GT	509	72.66 ± 0.99 ^a^	94.98 ± 0.63 ^a^	2.16 ± 0.13 ^a^	1.94 ± 0.28 ^a^	7.83 ± 0.63 ^a^	24 ± 0.05 ^a^
5 mg/mL	566	71 ± 0.19 ^a^	94.92 ± 0.57 ^a^	2.18 ± 0.16 ^a^	1.96 ± 0.22 ^a^	7.01 ± 0.76 ^a^	25 ± 0.99 ^a^
**48 h**							
1.25 mg/mL GT	516	74.25 ± 0.25 ^a^	95.03 ± 0.47 ^a^	2.01 ± 0.16 ^a^	2.19 ± 0.16 ^a^	7.01 ± 0.51 ^a^	22 ± 0.99 ^a^
2.5 mg/mL GT	569	73.66 ± 0.91 ^a^	94.11 ± 0.06 ^a^	2.05 ± 0.11 ^a^	1.22 ± 0.18 ^a^	7.99 ± 0.12 ^a^	24 ± 0.75 ^a^
5 mg/mL	568	74 ± 0.28 ^a^	94.92 ± 0.23 ^a^	2.22 ± 0.02 ^a^	1.98 ± 0.32 ^a^	7.91 ± 0.75 ^a^	25 ± 0.19 ^a^

Different letters in the same columns show the statistically significant among treatment groups (*p* ≤ 0.05). MMS = Methyl methanesulphonate; CCN = counting cell numbers; SD = Standard deviation.

**Table 6 biomedicines-11-01959-t006:** Effect of ampicillin and gel (GA) on mitotic and phase index in *A. cepa* roots and impact on DNA damage at different concentrations.

Concentration (ppm)	CCN	MI ± SD	Phase Index (%) ± SD				DNA Damage (Mean ± SD)
			Prophase	Metaphase	Anaphase	Telophase	
**24 h**							
Control	576	76.21 ± 0.92 ^a^	99.98 ± 0.61 ^a^	2.19 ± 0.11 ^a^	2.68 ± 0.29 ^a^	7.91 ± 0.89 ^a^	12 ± 0.19 ^a^
MMS	574	59.15 ± 0.67 ^b^	86.13 ± 0.46 ^b^	2.01 ± 0.26 ^b^	1.3 ± 0.11 ^b^	6.15 ± 0.15 ^b^	123 ± 0.75 ^b^
1.25 mg/mL GA	577	72.11 ± 0.75 ^a^	95.43 ± 0.46 ^a^	2.13 ± 0.10 ^a^	2.99 ± 0.19 ^a^	7.85 ± 0.15 ^a^	19 ± 0.29 ^a^
2.5 mg/mL GA	507	72.19 ± 0.99 ^a^	94.99 ± 0.13 ^a^	2.06 ± 0.03 ^a^	1.98 ± 0.08 ^a^	7.83 ± 0.60 ^a^	20 ± 0.22 ^a^
5 mg/mL GA	545	71.72 ± 0.91 ^a^	93.15 ± 0.76 ^a^	2.07 ± 0.17 ^a^	1.97 ± 0.19 ^a^	7.98 ± 0.62 ^a^	19 ± 0.15 ^a^
**48 h**							
1.25 mg/mL GA	518	73.86 ± 0.99 ^a^	93.92 ± 0.57 ^a^	2.18 ± 0.16 ^a^	1.96 ± 0.82 ^a^	7.21 ± 0.76 ^a^	20 ± 0.22 ^a^
2.5 mg/mL GA	519	74.15 ± 0.25 ^a^	94.03 ± 0.47 ^a^	2.87 ± 0.19 ^a^	2.99 ± 0.86 ^a^	7.91 ± 0.71 ^a^	20 ± 1.99 ^a^
5 mg/mL GA	569	73.16 ± 0.91 ^a^	94.11 ± 0.06 ^a^	2.85 ± 0.19 ^a^	1.12 ± 0.88 ^a^	7.97 ± 0.19 ^a^	20 ± 1.23 ^a^

Different letters in the same columns show the statistically significant among treatment groups (*p* ≤ 0.05). MMS = Methyl methanesulphonate; CCN = counting cell numbers; SD = Standard deviation.

**Table 7 biomedicines-11-01959-t007:** Effect of gel, magnesium oxide, and tylosin (GMT) on mitotic and phase index in *A. cepa* roots and impact on DNA damage at different concentrations.

Concentration (ppm)	CCN	MI ± SD	Phase Index (%) ± SD				DNA Damage (Mean ± SD)
			Prophase	Metaphase	Anaphase	Telophase	
**24 h**							
Control	586	78.22 ± 0.12 ^a^	98.09 ± 0.72 ^a^	2.02 ± 0.12 ^a^	2.62 ± 0.28 ^a^	7.98 ± 0.19 ^a^	12 ± 0.99 ^a^
MMS	575	56.15 ± 0.67 ^b^	84.13 ± 0.06 ^b^	2.01 ± 0.16 ^b^	1.2 ± 0.01 ^b^	5.88 ± 0.87 ^b^	122 ± 0.75 ^b^
1.25 mg/mL GMT	577	76.12 ± 0.81 ^a^	96.13 ± 0.36 ^a^	2.99 ± 0.12 ^a^	2.99 ± 0.17 ^a^	7.65 ± 0.92 ^a^	
2.5 mg/mL GMT	550	75.09 ± 0.19 ^a^	94.19 ± 0.12 ^a^	2.09 ± 0.01 ^a^	1.68 ± 0.01 ^a^	7.13 ± 0.05 ^a^	25 ± 0.59 ^a^
5 mg/mL GMT	545	75.02 ± 0.92 ^a^	94.10 ± 0.45 ^a^	2.06 ± 0.16 ^a^	1.87 ± 0.18 ^a^	7.25 ± 0.22 ^a^	27 ± 0.09 ^a^
**48 h**							
1.25 mg/mL GMT	545	75.16 ± 0.19 ^a^	93.92 ± 0.06 ^a^	2.08 ± 0.26 ^a^	1.86 ± 0.82 ^a^	6.91 ± 0.06 ^a^	26 ± 0.02 ^a^
2.5 mg/mL GMT	559	74.15 ± 0.25 ^a^	93.23 ± 0.17 ^a^	2.07 ± 0.79 ^a^	1.09 ± 0.86 ^a^	6.01 ± 0.22 ^a^	26 ± 0.01 ^a^
5 mg/mL GMT	550	73.16 ± 0.91 ^a^	93.95 ± 0.06 ^a^	2.15 ± 0.02 ^a^	1.02 ± 0.88 ^a^	6.07 ± 0.35 ^a^	25 ± 0.12 ^a^

Different letters in the same columns show the statistically significant among treatment groups (*p* ≤ 0.05). MMS = Methyl methanesulphonate; CCN = counting cell numbers; SD = Standard deviation.

**Table 8 biomedicines-11-01959-t008:** Percentage variations in ZOI of individual drugs compared with composites.

Antibiotics/Nanoparticle	Combinations	Mean ± SD	Percentage (%) Variation
Tylosin	Alone	14.000 ± 1.000 ^c^	-
	GT	33.67 ± 5.13 ^a^	58.42%
	GMT	26.00 ± 3.46 ^ab^	46.15%
Cefoxitin	Alone	18.00 ± 5.29 ^a^	-
	GC	30.33 ± 1.528 ^a^	40.65%
	GMC	25.00 ± 3.61 ^a^	40%
Ampicillin	Alone	24.00 ± 7.00 ^a^	-
	GA	30.67 ± 8.08 ^a^	21.75%
	GMA	33.67 ± 5.69 ^a^	28.72%
	MA	23.33 ± 6.66 ^a^	−2.87%
Magnesium oxide	Alone	18.67 ± 0.577 ^b^	-
	GMC	25.00 ± 3.61 ^ab^	25.32%
	GMT	26.00 ± 3.46 ^ab^	28.19%
	GMA	33.67 ± 5.69 ^a^	44.55%
	MG	22.67 ± 5.03 ^ab^	17.65%

Gel, MgO, and ampicillin (GMA); gel and ampicillin (GA); GC = gel and cefoxitin; gel and tylosin (GT); gel, MgO, and cefoxitin (GMC); gel, MgO, and tylosin (GMT); MgO and gel (MG). Different letters in the same columns for each antibiotic/nanoparticle group show the statistically significant among treatment groups (*p* ≤ 0.05). Percentage variation (%) = (ZOI produced by preparation used in combination-ZOI produced by preparation used alone)/(ZOI produced by preparation used in combination) × 100.

**Table 9 biomedicines-11-01959-t009:** Minimum inhibitory concentration (µg/mL) among different composites.

Drug	4 h	8 h	12 h	16 h	20 h	24 h
GMA	625.0 ± 0.0 ^a^	312.5 ± 0.0 ^b^	208.3 ± 90.2 ^a^	65.1 ± 22.6 ^b^	16.28 ± 5.64 ^c^	9.766 ± 0.000 ^b^
GA	833 ± 361 ^a^	625.0 ± 0.0 ^ab^	365 ± 239 ^a^	130.2 ± 45.1 ^ab^	52.1 ± 22.6 ^bc^	26.04 ± 11.28 ^ab^
GT	1042 ± 361 ^a^	833 ± 361 ^a^	521 ± 180 ^a^	104.2 ± 45.1 ^ab^	52.1 ± 22.6 ^bc^	13.02 ± 5.64 ^b^
GMC	625.0 ± 0.0 ^a^	312.5 ± 0.0 ^b^	156.3 ± 0.0 ^a^	130.2 ± 45.1 ^ab^	32.55 ± 11.28 ^bc^	19.53 ± 0.00 ^b^
GMT	625.0 ± 0.0 ^a^	312.5 ± 0.0 ^b^	156.3 ± 0.0 ^a^	78.13 ± 0.00 ^b^	39.06 ± 0.00 ^bc^	26.04 ± 11.28 ^ab^
MG	1042 ± 361 ^a^	521 ± 180 ^ab^	365 ± 239 ^a^	182.3 ± 119.3 ^ab^	65.1 ± 22.6 ^b^	39.06 ± 0.00 ^a^
GC	833 ± 361 ^a^	625.0 ± 0.0 ^ab^	312.5 ± 0.0 ^a^	260.4 ± 90.2 ^a^	156.3 ± 0.0 ^a^	39.06 ± 0.00 ^a^

Different superscripts placed on mean ± SD values within the column show significant difference (*p* < 0.05). Gel, MgO, and ampicillin (GMA); gel and ampicillin (GA); gel and tylosin (GT); gel and cefoxitin (GC); gel, MgO, cefoxitin (GMC); gel, MgO, and tylosin (GMT); MgO and gel (MG).

## Data Availability

Data is available in the manuscript.

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
