# Peer review of "Sodium Alginate-Based MgO Nanoparticles Coupled Antibiotics as Safe and Effective Antimicrobial Candidates against Staphylococcus aureus of Houbara Bustard Birds"

_biomedicines, 2023, doi:10.3390/biomedicines11071959_

Round 1

Reviewer 1 Report

The paper is interesting and falls within the scope of the journal. The experimental work is described in some detail to allow reproducibility  and statistical analysis is provided. Figure 5 and in particular y-axis is hard to read and should be replotted.  

Author Response

The paper is interesting and falls within the scope of the journal.

Answer: Thank you for appreciation our research work

The experimental work is described in some detail to allow reproducibility and statistical analysis is provided. Figure 5 and in particular y-axis is hard to read and should be replotted.  

Answer: We have revised figrue 5. In current form of figure 5, we have increased visibility by increasing interval on y-axis from 50 to 100.

We appreciate your comment in this regard.

Reviewer 2 Report

This study investigated the cytotoxicity and antibacterial potential of composites of magnesium oxide nanoparticles and antibiotics stabilized in sodium alginate gel against multi-drug resistant Staphylococcus aureus. The composites demonstrated enhanced mitotic indices and lower DNA damage compared to individual components. They also showed increased zone of inhibition in well diffusion trials and lower minimum inhibitory concentrations in broth microdilution assays. The study concludes that sodium alginate-stabilized MgO nanoparticles and antibiotics effectively combat multi-drug resistant S. aureus with negligible toxicity.

 Please state the novelty of the study

Please number the equations!

Please double check the paper for typos.

Figure 5. e and g – are unclear. Please replace!

The conclusion suggests further in vivo and field trials for dose standardization and product development. While this is a reasonable suggestion, it would be beneficial to discuss potential challenges or limitations that may arise during these trials.

Minor editing of English language required

Author Response

This study investigated the cytotoxicity and antibacterial potential of composites of magnesium oxide nanoparticles and antibiotics stabilized in sodium alginate gel against multi-drug resistant Staphylococcus aureus. The composites demonstrated enhanced mitotic indices and lower DNA damage compared to individual components. They also showed increased zone of inhibition in well diffusion trials and lower minimum inhibitory concentrations in broth microdilution assays. The study concludes that sodium alginate-stabilized MgO nanoparticles and antibiotics effectively combat multi-drug resistant S. aureus with negligible toxicity.

Answer: We are appreciating your analysis and have tried best to revise manuscript.

 Please state the novelty of the study

Answer: Statement of novel has been briefly added.

Please number the equations!

Answer: Thank you for pointing out this overlooked point. We have given numbers to the equations as suggested

Please double check the paper for typos.

 Answer: We have tried to eliminate typos as suggested

Figure 5. e and g – are unclear. Please replace!

 Answer: We agree with your comment and have revised figure 5. Now the values are easily readable as we extended intervals at y axis from 50 to 100.

The conclusion suggests further in vivo and field trials for dose standardization and product development. While this is a reasonable suggestion, it would be beneficial to discuss potential challenges or limitations that may arise during these trials.

Answer: You suggestion is highly appreciable, and it is sort of very important research question. We have added challenges and limitations as “Extensive studies are required to validate outcomes of therapeutic and toxicity trials, refinement of dose regimens is another challenge, binding ability of drugs with nanoparticles poses a challenge, and stability of nanoparticle structure while using in-vivo trials. However, these trials may strengthen strategies to counter antimicrobial resistance and infection at satisfactory level.”

Minor editing of English language required.

Answer: We have tried our level best to proofread manuscript for any mistake.